# Identification and Validation of the Prognostic Panel in Clear Cell Renal Cell Carcinoma Based on Resting Mast Cells for Prediction of Distant Metastasis and Immunotherapy Response

**DOI:** 10.3390/cells12010180

**Published:** 2023-01-01

**Authors:** Yang Su, Tianxiang Zhang, Jinsen Lu, Lei Qian, Yang Fei, Li Zhang, Song Fan, Jun Zhou, Jieqiong Tang, Haige Chen, Chaozhao Liang

**Affiliations:** 1Department of Urology, The First Affiliated Hospital of Anhui Medical University, Hefei 230031, China; 2Anhui Province Key Laboratory of Genitourinary Diseases, Anhui Medical University, Hefei 230031, China; 3The Institute of Urology, Anhui Medical University, Hefei 230031, China; 4Department of Urology, Renji Hospital, School of Medicine, Shanghai Jiao Tong University, Shanghai 200120, China; 5State Key Laboratory of Oncogenes and Related Genes, Shanghai Cancer Institute, Renji Hospital, School of Medicine, Shanghai Jiao Tong University, Shanghai 200120, China; 6Nuffield Department of Orthopaedics, Rheumatology and Musculoskeletal Sciences, University of Oxford, Oxford OX3 7LD, UK; 7Department of Cardiology, Chuzhou Hospital Affiliated to Anhui Medical University, Chuzhou 239000, China

**Keywords:** clear cell renal cell carcinoma, distant metastasis, immunotherapy, prognostic panel, resting mast cells, tumor microenvironment

## Abstract

Clear cell renal cell carcinoma (ccRCC) has a high metastatic rate, and its incidence and mortality are still rising. The aim of this study was to identify the key tumor-infiltrating immune cells (TIICs) affecting the distant metastasis and prognosis of patients with ccRCC and to construct a relevant prognostic panel to predict immunotherapy response. Based on ccRCC bulk RNA sequencing data, resting mast cells (RMCs) were screened and verified using the CIBERSORT algorithm, survival analysis, and expression analysis. Distant metastasis-associated genes were identified using single-cell RNA sequencing data. Subsequently, a three-gene (*CFB*, *PPP1R18,* and *TOM1L1*) panel with superior distant metastatic and prognostic performance was established and validated, which stratified patients into high- and low-risk groups. The high-risk group exhibited lower infiltration of RMCs, higher tumor mutation burden (TMB), and worse prognosis. Therapeutically, the high-risk group was more sensitive to anti-PD-1 and anti-CTLA-4 immunotherapy, whereas the low-risk group displayed a better response to anti-PD-L1 immunotherapy. Furthermore, two immune clusters revealing distinct immune, clinical, and prognosis heterogeneity were distinguished. Immunohistochemistry of ccRCC samples verified the expression patterns of the three key genes. Collectively, the prognostic panel based on RMCs is able to predict distant metastasis and immunotherapy response in patients with ccRCC, providing new insight for the treatment of advanced ccRCC.

## 1. Introduction

Renal cell carcinoma (RCC) accounts for nearly 4% of all newly diagnosed cancers and 2.5% of cancer deaths worldwide [1]. Clear cell renal cell carcinoma (ccRCC) accounts for approximately 80–90% of RCC, leading to death in most patients [2]. Globally, the incidence and mortality of ccRCC have increased in the past few decades, resulting in a heavy burden on the healthcare system [3]. Due to the lack of specific clinical symptoms and diagnostic methods, 20–25% of patients with ccRCC present with metastatic disease at diagnosis, and their 5-year survival rate is less than 10% [4,5]. Traditional radiotherapy and chemotherapy have poor curative effects on ccRCC. Surgery is the most effective treatment for localized RCC, yet uncontrolled distant metastasis and death still develop in 25-40% of patients after surgery [4]. Targeted therapeutic agents, such as vascular endothelial growth factor receptor inhibitors and mTOR inhibitors, can improve the survival of patients with ccRCC [6]. However, many patients with ccRCC have no targetable mutations, and targeted therapy fails in patients with advanced ccRCC because of target tolerance [7]. Recently, the clinical application of novel immunotherapies including cytokines, immune checkpoint inhibitors (ICIs), and adoptive cells has dramatically improved the outcomes of patients with metastatic ccRCC [4,8]. However, reliable biomarkers that predict efficacy in patients with ccRCC are currently lacking [9]. Therefore, finding new therapeutic targets and constructing a more powerful model to guide the treatment of patients with advanced ccRCC are urgently needed.

Solid tumors consist of cancer cells surrounded by a complex tumor microenvironment (TME), including immune cells, stromal cells, cytokines, and extracellular matrix molecules [10]. Accumulating evidence has revealed that the TME is of great significance in tumor growth and progression [11]. Among these noncancer cells in the TME, tumor-infiltrating immune cells (TIICs) play a central role in predicting the metastasis and mortality of various tumors [12]. Furthermore, dysfunction of TIICs is closely associated with prognosis and response to immunotherapy in cancer patients [13]. However, compared with other carcinomas, the differing distribution and functional effects of specific TIICs in metastatic ccRCC have not yet been elucidated [14]. Traditional methods for exploring cell heterogeneity in the TME, including immunohistochemistry (IHC) and flow cytometry, can lead to tissue disaggregation and cell loss, making the results unreliable. As an emerging computational approach, CIBERSORT avoids the possible dissociation of cells and types of surface markers, making it more accurate for the quantification of different types of TIICs [15]. With the advancement of next-generation sequencing, RNA sequencing (RNA-seq) of bulk tissue has improved our understanding of tumor occurrence and development [16]. However, considering the complex interaction of tumor cells and the associated nontumor constituents in the TME, RNA-seq data based on the average expression levels of bulk tumor cells have limitations in terms of understanding the heterogeneity of metastatic tumors. Emerging single-cell technologies, which provide a high-resolution tumor immune landscape, could exert a complementary effect in principle [17].

Although some studies have constructed models in the TME to predict ccRCC prognosis, a prognostic panel based on the key TIICs associated with distant metastasis has not been explored [18,19]. Therefore, in this study, we aimed to identify the key TIICs that play a central role in ccRCC distant metastasis and prognosis. Subsequently, by integrating bulk and single-cell RNA-seq data, a signature based on the identified TIICs that can independently predict prognosis and distant metastasis in ccRCC was established and validated in ccRCC tissues using IHC. Furthermore, two immune subtypes revealing distinct immune, clinical, and prognosis heterogeneity were identified. We also systematically investigated the relationship of the prognostic panel with tumor mutation burden (TMB), immune checkpoints, and ICI immunotherapy responses in patients with ccRCC.

## 2. Materials and Methods

### 2.1. Acquisition of Bulk RNA-Seq Data

Data for 539 human ccRCC tissue samples and 72 normal tissue samples were downloaded from the Cancer Genome Atlas (TCGA, https://portal.gdc.cancer.gov, accessed on 15 January 2022), including RNA-seq data and corresponding clinical information. We also downloaded the expression profiles of 91 ccRCC and 45 normal kidney tissues from the International Cancer Genome Consortium (ICGC, https://icgc.org/, accessed on 25 January 2022). Normalized microarray gene expression datasets GSE53757, GSE66272, and GSE126964 were available from the Gene Expression Omnibus (GEO, http://www.ncbi.nlm.nih.gov/geo/, accessed on 12 March 2022) according to the following criteria: (1) pathological diagnosis of ccRCC; (2) samples were all derived from human kidney tissues; and (3) the number of samples was no less than 50. TCGA and ICGC datasets were used to screen for key differentially expressed genes (DEGs). We then utilized the three cohorts obtained from the GEO database as external validation datasets. An overview of this study is presented in Figure 1.

### 2.2. Data Processing and DEG Screening

Transcriptome counts were normalized using the R package “limma” [20], and the gene expression values from TCGA and ICGC datasets were transformed as log2 (x + 1) counts. The R package “DEseq2” [21] was used to screen DEGs between ccRCC and normal kidney samples. The screening criteria for DEGs was set to |log2FoldChange| > 1, with an adjusted *p*-value < 0.05. Venn diagrams were drawn to obtain the intersection of DEGs from TCGA and ICGC using a web tool (http://bioinformatics.psb.ugent.be/webtools/Venn/, accessed on 25 March 2022). Volcano plots were acquired based on “ImageGP” and “EnhancedVolcano” packages, and heat maps were generated using the R package “pheatmap”.

### 2.3. Acquisition and Analysis of Single-Cell RNA Sequencing (scRNA-seq) Data for Distant Metastatic Sites in ccRCC

The scRNA-seq profiles of 121 ccRCC cells produced using the Illumina HiSeq 2500 platform were obtained from the GSE73121 dataset in the GEO database. Then, 85 tumor cells isolated from patient-derived xenografts of metastatic RCC (PDX-mRCC) and patient-derived xenografts of primary RCC (PDX-pRCC) were extracted and processed using the “Seurat” package. First, we filtered cells with fewer than 1000 detected gene numbers and excluded genes with fewer than three detected cells. The PercentageFeatureSet function was used to evaluate the percentage of mitochondrial genes, and cells with a mitochondrial gene content of less than 5% were screened. The correlation between sequencing depth and gene expression counts was calculated using the FeatureScatter function. Subsequently, data normalization was performed using the NormalizeData method, and the top 2000 highly variable genes were screened using the FindVariableFeatures function. After principal component analysis (PCA), significantly available dimensions with an estimated *p*-value < 0.05 were identified, and the t-distributed stochastic neighbor-embedding (t-SNE) algorithm was employed for dimension optimization. Moreover, the FindAllMarkers function was utilized to identify the significant marker genes in each cell cluster.

### 2.4. Assessment of TIICs Using CIBERSORT

The “CIBERSORT” R package [15], an algorithm used to characterize the immune cell composition of complex samples based on a particular gene expression profile, was utilized to estimate the infiltrating fraction of TIICs in ccRCC samples obtained from TCGA. LM22, a leukocyte gene matrix containing 547 specific genes that distinguish 22 human infiltrating immune cell subpopulations, including seven T-cell types, naive and memory B cells, plasma cells, natural killer (NK) cells, and myeloid subsets, served as a reference expression signature.

### 2.5. Weighted Gene Coexpression Network Analysis (WGCNA)

The R package “WGCNA” was used to employ WGCNA, a comprehensive algorithm that utilizes a gene expression matrix to construct a scale-free network [22]. First, a sample tree was built to check all samples for outliers using the goodSamplesGenes function. The best soft threshold parameter was then screened to generate an adjacency matrix by calculating a group of soft threshold powers (range: 1–20). Subsequently, a topological overlap measure was performed to ensure the non-scaling of the network. In the scale-free network, genes with high correlations were classified into the same module to construct a cluster dendrogram. The parameters were set as follows: TOMType = unsigned, minModuleSize = 50, deepSplit = 2, and mergeCut Height = 0.15. Finally, we linked the coexpressed modules with key TIICs using Pearson’s correlation method. The module with the highest absolute correlation coefficient value was filtered for subsequent analysis.

### 2.6. Gene Functional Enrichment Analysis

The R package “clusterProfiler” [23] was used to perform gene ontology (GO) function analysis and *Kyoto Encyclopedia of Genes and Genomes* (KEGG) pathway enrichment analysis. The top 20 terms were filtered according to an adjusted *p*-value < 0.05. Metascape (http://pantherdb.org, accessed on 23 June 2022), a comprehensive database that plays an important role in the annotation and visualization of the gene list, was used for pathway analysis and protein–protein interaction (PPI) analysis [24].

### 2.7. Correlation Analysis between Mast Cell Markers and Distant Metastasis-Associated Resting Mast-Cell-Sensitive Genes (DMA-RMSGs)

We compared the same elements in the key resting mast cell (RMC)-related module and distant metastasis-associated marker genes (DMAGs) to obtain DMA-RMSGs. Marker genes of mast cells were downloaded from the Panglao database (https://panglaodb.se/, accessed on 15 July 2022). The search criteria were initially limited to human donor tissue. Pearson’s algorithm was used to examine the association between DMA-RMSGs and mast cell markers. Furthermore, correlation analysis was performed using the LabeledHeatmap function in the R package WGCNA.

### 2.8. Key DMA-RMSG Screening and Prognostic Model Establishment

DMA-RMSGs with prognostic potential were first screened using univariate Cox regression analysis (*p* < 0.05). Then, Kaplan–Meier analysis was utilized to validate the prognostic performance of DMA-RMSGs based on the “survival” package. To find the hub prognostic DMA-RMSGs, a least absolute shrinkage and selection operator (LASSO) regression analysis was conducted using the R package “glmnet”. Moreover, the Akaike information criterion (AIC), which reflects the goodness of fit among different prognostic factors, was calculated and compared to enhance the prediction accuracy of the model. Furthermore, multivariate Cox regression analysis was used to identify the independent prognostic DMA-RMSGs (*p* < 0.05). Then, the differences in the expression levels of key DMA-RMSGs between distant metastasis and other important clinical phenotypes were illustrated using the R package “ggpubr”. Based on the screened independent prognostic factors and the regression coefficient (β) obtained from the multivariate Cox regression analysis, we constructed a risk score model as follows:$$\mathrm{Risk}\;\mathrm{score}=\sum_{i=1}^{n}{\mathrm{Coef}}_{i}\times {\mathrm{Exp}}_{i}$$

Additionally, the median risk score was calculated to classify patients with ccRCC into low- and high-risk groups.

### 2.9. Consensus Clustering Analysis

Based on prognosis-related DMA-RMSGs, unsupervised clustering analysis was conducted to identify the stable immune subtypes in ccRCC. The number of clusters and their stability were assessed by the consensus clustering method. The clustering sets ranged from 2 to 9, and the optimal stratification was determined by evaluating the consistency array and the accumulative distribution. The R package “ConsensusClusterPlus” was utilized to conduct 1000 repetitions to ensure accuracy of the divided subtypes. Additionally, PCA and t-SNE algorithms were implemented to validate the distinct distribution of the DMA-RMSG clusters.

### 2.10. Evaluation and Verification of the Prognostic Model

To assess the predictive accuracy of the risk score model, a time-dependent receiver operating characteristic (ROC) curve analysis was performed. The area under the curve (AUC) at different cutoff times was calculated using the “survival ROC” package. Furthermore, decision curve analysis (DCA) was conducted using the R package “ggDCA” to verify the practicality and reliability of the signature. The “scatterplot3d” package was utilized to conduct PCA between different risk groups. Then, we drew a heat map to illustrate the relationship between the risk score and clinical traits using the “ComplexHeatmap” package. Validation of the expression patterns and disease-free survival (DFS) of key DMA-RMSGs in the prognostic panel was determined using several ccRCC databases, such as GEO (GSE53757, GSE66272, and GSE126964), ICGC, and GEPIA (http://gepia.cancer-pku.cn/, accessed on 25 July 2022). The prognostic performance of the key DMA-MRSGs was validated using the Kaplan–Meier plotter database (http://kmplot.com/analysis/, accessed on 27 July 2022). Furthermore, the TIMER2.0 database (http://timer.comp-genomics.org/, accessed on 2 August 2022) was used to conduct a pan-cancer analysis based on the key DMA-RMSGs [25].

### 2.11. Gene Set Enrichment Analysis (GSEA)

To explore the potential signaling pathways in the high- and low-risk groups, GSEA (http://software.broadinstitute.org/gsea/index.jsp, accessed on 15 July 2022) was conducted based on the TCGA ccRCC cohort. The annotated gene set list “c2.cp.kegg.v7.5.symbols.gmt” was used as a reference gene set. After performing 1000 permutations, a *p*-value < 0.05 and a false-discovery rate < 0.25 were regarded as statistically significant. We then used the EnrichmentMap package [26] in Cytoscape3.7.1 to visualize the enriched signaling pathways.

### 2.12. Evaluation of ICI Immunotherapy Efficacy

The Cancer Immunome Atlas (TICA; https://tcia.at, accessed on 17 July 2022) is a database that characterizes intratumoral immune landscapes and cancer antigenomes from 20 solid cancers. The immuno scores of patients with ccRCC who received anticytotoxic T lymphocyte antigen-4 (CTLA-4) and anti-programmed cell death protein 1 (anti-PD-1) treatments were downloaded from the TICA database. Moreover, the transcriptomic information and matching clinical data of patients who received anti-PD-L1 treatment were acquired from IMvigor210 (http://research-pub.gene.com/IMvigor210CoreBiologies, accessed on 28 July 2022).

### 2.13. Nomogram Construction and Evaluation

After performing univariate and multivariate Cox regression analyses of the risk score and various clinicopathological factors (distant metastasis, age, sex, race, stage, grade, and N stage), the independent prognostic factors were identified. Subsequently, to evaluate the prognosis of patients with ccRCC, the “rms” R package was used to build a nomogram based on the selected independent prognostic factors. The prognostic performance of the nomogram was evaluated based on the time-dependent ROC curves using the “timeROC” R package. Additionally, calibration curves were generated for different years to measure the accuracy of the nomogram.

### 2.14. Correlation Analysis between the Risk Score Model and TMB

TMB refers to the total number of somatic gene-coding errors, base substitutions, insertions, and deletions detected per million bases. Gene mutation data of patients with ccRCC were obtained from the TCGA database. TMB was calculated as the number of variants/length of the exons (38 MB) using the “maftools” package [27]. Survival analysis was performed to detect the relationship between TMB alone or in combination with the risk model and overall survival (OS) of patients with ccRCC using the “survminer” R package.

### 2.15. Autophagy-Related Genes (ATGs) Associated with the DMA-RMSG-Based Signature

A total of 222 ATGs were collected from the human autophagy database (HADb, http://www.autophagy.lu/, accessed on 9 August 2022), which collects autophagy-related information from papers in PubMed and other biological databases [28]. We utilized correlation analysis to screen the underlying ATGs by which the prognostic panel was potentially regulated, and the ATG with the highest correlation coefficient was identified as the key ATG.

### 2.16. Tissue Specimen Collection

Resection and biopsy were performed to collect ccRCC and adjacent tissue specimens at the Renji Hospital (Shanghai, China). The inclusion criteria for tissue samples are as follows: (1) pathological diagnosis of ccRCC, (2) no other malignancy except ccRCC, (3) conformation with indications for radical or partial nephrectomy, and (4) no chemotherapy or radiotherapy before surgery. Two senior pathologists from our institution confirmed the pathological diagnosis of ccRCC after surgery. All participants provided written informed consent, and the protocol was approved by the Ethics Committee of Human Research of the Hospital.

### 2.17. IHC

Tissues were embedded in paraffin and sectioned after surgery. The tissue sections were dewaxed, rehydrated, and autoclaved at 110 °C for 5 min for antigen retrieval in citric acid buffer (0.01 M, pH 6.0), incubated with 3% H_2_O_2_ (SP9000; Sangon Biotech Shanghai Co., Ltd.; Shanghai, China) solution for 15 min, and blocked with 10% bovine serum albumin. Subsequently, the slides were incubated overnight at 4 °C with anti-CFB (1:2000, 20103-1-AP; Proteintech; Wuhan, China), anti-PPP1R18 (1:1000, DF6417; Affinity; Changzhou, China), and anti-TOM1L1 (1:50, DF7113; Affinity; Changzhou, China) antibodies. The primary antibody was then washed off with PBS, and biotinylated goat anti-rabbit IgG (1:200) was added for 2 h at room temperature. Eventually, the immune complexes were detected using DAB (ZLI-0918; ZSBio; Beijing, China), and nuclei were counterstained with hematoxylin.

### 2.18. Statistical Analysis

Data analysis and visualization were performed using R software (version 4.2.1, https://www.rstudio.com/, accessed on 3 January 2022) and GraphPad 9.0 (GraphPad Software, San Diego, CA, USA). The significance of differences in the expression of genes among pathological stages or histological grades was evaluated using one-way ANOVA. Student’s t-test was used to compare two variables, such as expression data from distant metastasis tissues and nondistant metastasis tissues or high- and low-risk patients. Survival analysis was conducted using the R package “survival”. Differences were considered statistically significant at *p* < 0.05.

## 3. Results

### 3.1. Screening for Key TIICs

We evaluated the landscape of 22 immune cell infiltrations in ccRCC tissues. The infiltrating cell proportions of each ccRCC sample are shown in Figure 2A. After ranking the infiltration levels of the 22 TIICs (Figure 2B), the top ten TIICs were filtered for the next analysis. Subsequently, we investigated the infiltration of the top ten TIICs between distant and nondistant metastasis samples in ccRCC (Figure 2C). Compared with nondistant metastasis tissues, RMCs and resting NK cells showed significantly lower infiltration in distant metastasis tissues, whereas CD8 T cells showed significantly higher infiltration. To explore the prognostic potential of the ten TIICs, an OS analysis was performed in the TCGA dataset (Figure 2D, Appendix A). The results indicated that RMCs were significantly related to favorable OS in patients with ccRCC (*p* < 0.001), whereas plasma cells were significantly associated with poor OS (*p* = 0.019). Additionally, patients with higher infiltration of RMCs also presented better progression-free survival (PFS) (*p* = 0.005, Figure 2E). Taken together, RMCs were identified as the key TIICs. Moreover, compared with normal kidney tissues, ccRCC tissues contained fewer RMCs (Figure 2F). Furthermore, the infiltration of RMCs significantly decreased with increasing T stage, pathological stage, and histological grade (*p* < 0.001, Figure 2G–I).

### 3.2. Identification of DEGs

Through differential expression analysis between ccRCC samples and normal adjacent samples available in the TCGA and ICGC datasets, we separately identified 11,912 (4029 downregulated and 7883 upregulated) and 10,574 (4281 downregulated and 6293 upregulated) DEGs. Volcano plots were constructed to describe the distribution of each DEG in TCGA and ICGC (Figure 3A,B). After comparing these DEGs between TCGA and ICGC, 6550 identical DEGs were screened, including 2590 downregulated and 3960 upregulated transcripts (Figure 3C,D, Appendix A). The expression patterns of DEGs among various clinical traits are shown using a heat map in Figure 3E.

### 3.3. WGCNA Identified RMSGs

To identify the key module associated with RMCs, we performed WGCNA of the screened DEGs. First, by performing a hierarchical clustering analysis of all ccRCC samples in TCGA, three samples with a distance greater than 250 (TCGA-CJ-4642) or less than 50 (TCGA-BP-4994, TCGA-BP-4995) were removed (Appendix A). The soft thresholding value was set as “4,” and a topological matrix with non-scale features (scale-free *R*^2^ = 0.88) was obtained (Appendix A). After placing DEGs with similar expression patterns into the same modules, 12 independent coexpressed modules were acquired (Figure 4A, Appendix A). The clustering tree and heat map of the module eigengenes showed that the DEGs in each module were independent of each other (Figure 4B). Moreover, the topologically overlapping heat map demonstrated a high degree of topological overlap between the coexpressed modules (Appendix A). Finally, we calculated the Pearson’s correlation coefficient between the coexpressed modules and ten TIICs (Figure 4C). Genes in the blue module that had the highest correlation coefficient with the key RMCs (*R* = −0.43, *p* = 3 × 10^−25^) were identified as RMSGs. The scatter plot of module membership vs. gene significance also revealed that RMSGs were highly associated with RMCs (Figure 4D, cor = 0.75, *p* = 1.3 × 10^−127^).

### 3.4. Functional Annotation of the RMSGs

Functional enrichment analysis was conducted to explore the potential functions and pathways of RMSGs (Appendix A). KEGG pathway enrichment analysis revealed that the RMSGs were mainly involved in immune and inflammatory pathways, including PD-L1 expression and the tPD-1 checkpoint pathway in cancer, the T-cell receptor signaling pathway, NK cell-mediated cytotoxicity, cytokine–cytokine receptor interaction, and the chemokine signaling pathway (Figure 5A). Consistent with the KEGG results, the top 20 enriched GO terms are shown in Figure 5B–D, including activation of immune response, regulation of T-cell activation, and the antigen receptor-mediated signaling pathway in the biological process; the T-cell receptor complex, immunoglobulin complex, and inflammasome complex in the cell component; and immunoglobulin receptor binding, antigen binding, and cytokine activity in molecular function. Furthermore, highly concordant results for diverse immune-related pathways and their interactions were obtained after enrichment analysis using the Metascape algorithm (Figure 5E). Moreover, PPI analysis revealed the underlying regulatory network among RMSGs (Figure 5F).

### 3.5. Quality Control and Normalization of scRNA-seq Profiling

To improve the accuracy of our analysis, quality control and normalization were performed in the scRNA-seq dataset GSE73121. First, we conducted quality control of the 85 tumor cells, including the scale of the detected gene numbers, sequencing counts, and mitochondrial gene sequences (Figure 6A). A significant positive correlation between nFeature-RNA and nCount-RNA was observed (*R* = 0.25, Figure 6B). Using variance analysis, 2000 highly variable DEGs across all cells were extracted for further analysis (Figure 6C).

### 3.6. Screening DMAGs Using Single-Cell Data

The PCA algorithm was used to identify the significantly correlated genes; the top 30 highly related genes in the first four components are displayed using dot plots and heat maps in Appendix A. Moreover, two independent distribution patterns of cell subpopulations based on principal component one and principal component two were observed (Figure 6D). After calculating the estimated *p*-value of other components, the principal components with a *p*-value < 0.05 were selected for the next analysis (Figure 6E). Additionally, after performing t-SNE on all samples, the cells were clustered into two subpopulations: primary tumor cells and distant metastatic tumor cells (Figure 6F). Subsequently, 583 DMAGs were screened using differential expression analysis between the two subpopulations. The distribution of the top 30 marker genes between the two subgroups is shown in Figure 6G.

### 3.7. Identification and Evaluation of DMA-RMSGs

After considering the intersection of DMAGs and RMSGs, 17 key genes were identified as DMA-RMSGs (Figure 6H). We then noted the independent expression patterns of DMA-RMSGs between the primary tumor cluster and the distant metastatic tumor cluster using a bubble plot (Figure 6I). Furthermore, according to the scatter plots of the t-SNE analysis, most DMA-RMSGs exhibited distinct expression patterns between the two clusters (Appendix A).

### 3.8. Screening DMA-RMSGs That Potentially Regulate Mast Cells

To further explore DMA-RMSGs that possibly regulate mast cells, a correlation analysis was conducted between the DMA-RMSGs and marker genes of mast cells (Appendix A), and an absolute value of the correlation coefficient of 0.5 was set as the cutoff criterion (Appendix A). As a result, IFI27 was excluded from the subsequent analysis. Heat maps were plotted to show the correlation among DMA-RMSGs before and after the removal of IFI27 (Figure 7A,B).

### 3.9. Identification and Evaluation of the Independent Prognostic DMA-RMSGs

After univariate Cox proportional hazard regression analysis, the 12 DMA-RMSGs that presented prognostic performance were filtered (Figure 7C). Kaplan–Meier survival analysis for OS validated the prognostic value of DMA-RMSGs. The result showed that the 12 selected DMA-RMSGs exhibited a significant prognosis value (*p* < 0.05), whereas the other four DMA-RMSGs did not (Appendix A). Subsequently, we performed LASSO regression analysis on the 12 prognostic DMA-RMSGs, and eight hub DMA-RMSGs were identified (Figure 7D,E). Moreover, AICs were calculated to evaluate the fit power of the eight hub DMA-RMSGs. The combination of the three key DMA-RMSGs (*CFB*, *PPP1R18*, and *TOM1L1*) had the lowest AIC value (1850.22) and was included in the next analysis (Appendix A). Subsequently, multivariate Cox regression analysis revealed that the three key DMA-RMSGs were independent prognostic factors in ccRCC (Figure 7F). We then evaluated the expression levels of the three prognostic DMA-RMSGs among several important clinical traits. Compared to nondistant metastatic ccRCC tissues and normal tissues, the expression patterns of *CFB* and *PPP1R18* were significantly higher in distant metastatic ccRCC samples and tumor samples, whereas the expression of *TOM1L1* was significantly lower (Figure 7G–L). Furthermore, the expression of *CFB* and *PPP1R18* increased with elevated pathological stage and histological grade, whereas the expression of *TOM1L1* decreased (Figure 7M–R). In addition, survival analysis showed that *CFB* and *PPP1R18* were significantly associated with poor DFS, whereas *TOM1L1* was significantly associated with favorable DFS (*p* < 0.01, Figure 7S–U).

### 3.10. Identification and Validation of DMA-RMSG Subtypes

The consensus clustering method was used to stratify patients with ccRCC based on the 12 prognosis-related DMA-RMSGs. After employing a set of k values, we found that a k value of two was the most stable parameter (Figure 8A), and the ccRCC patients were divided into two clusters (Figure 8B). Then, an independent distribution of the two DMA-RMSG subtypes was validated in both PCA and t-SNE analysis (Figure 8C,D). Furthermore, survival analysis demonstrated that the subtype-one patients displayed worse OS (*p* < 0.001, Figure 8E) and PFS (*p* < 0.001, Figure 8F). Among the 12 prognosis-related DMA-RMSGs, 9 DMA-RMSGs (*PPP1R18*, *EMP3*, *APOL2*, *GBP2*, *TGM2*, *RPS19*, *TAPBP*, *CFB*, and *BATF3*) linked with poor prognosis (HR > 1, *p* < 0.05) were upregulated in cluster one, whereas the other 3 DMA-RMSGs (*TOM1L1*, *CTH*, *EPCAM*) related to favorable prognosis (HR < 1, *p* < 0.05) were downregulated in cluster one (Figure 8G).

### 3.11. Clinical Characteristics and Immune Status between the Two Clusters

To further explore the potential clinical application of the two DMA-RMSG subtypes, we examined the relationship between the subtypes and various of clinical traits. The results showed that patients in cluster one had a higher proportion of distant metastasis, high stage, high grade, high TMB, and male sex compared to patients in cluster two (Figure 8H–L). However, there was no obvious difference in age between the two clusters (Figure 8M). Subsequently, the distributions of TIICs between the two clusters were also investigated. The patients in cluster two had higher infiltration of most TIICs (*p* < 0.001), including RMCs, M2 macrophages, monocytes, resting NK cells, and resting CD4 memory T cells, which could lead to a favorable prognosis of patients in cluster two (Figure 8N). Moreover, the results of TIIC infiltration obtained using algorithms showed a similar trend (Figure 8O).

### 3.12. Construction and Evaluation of the Risk Score Model Related to Distant Metastasis

According to the results of multivariate Cox regression analysis, three independent prognostic factors were selected to establish a prognostic panel as follows: risk score = (0.4662 × expression level of *PPP1R18*) + (−0.3206 × expression level of *TOM1L1*) + (0.1914 × expression level of *CFB*). The survival rate of low-risk patients was significantly higher than that of high-risk patients (Figure 9A). The risk signature that predicted OS after one, three, and five years showed AUC values of 0.752, 0.672, and 0.696, respectively, demonstrating the accuracy of our model in predicting ccRCC prognosis (Figure 9B). In addition, DCA revealed that the risk model had a good net benefit for OS compared with the key DMA-RMSGs alone (Figure 9C). The distribution plots revealed that high risk scores were associated with more deaths (Figure 9D,E). Furthermore, the risk scores of patients with distant metastasis ccRCC were significantly higher (*p* < 0.0001), demonstrating the stable power of our model in predicting distant metastasis in ccRCC (Figure 9F). PCA based on the prognostic panel displayed distinct distribution patterns, revealing remarkably different immune characteristics between the high- and low-risk groups (Figure 9G). Additionally, a heat map showed independent expression patterns of the three key DMA-RMSGs in high- and low-risk patients (Figure 9H). Finally, the heat map verified that the prognostic panel was significantly correlated with distant metastasis, gender, pathological stage, histological grade, T stage, and N stage (*p* < 0.05, Figure 9I).

### 3.13. Validation of the Prognostic Panel Based on the Key DMA-RMSGs

To validate the stability of our risk model, we evaluated the expression patterns of key DMA-RMSGs using other databases. In the three GEO datasets, the expression of *CFB* and *PPP1R18* was significantly higher in tumor samples, whereas that of *TOM1L1* was significantly lower (Appendix A). In the ICGC dataset, *CFB* and *PPP1R18* were significantly upregulated in tumor samples, whereas *TOM1L1* was significantly downregulated in tumor samples (Appendix A). Moreover, in GSE53757 and GEPIA datasets, the expression of *CFB* and *PPP1R18* increased with elevated pathological stage, whereas the expression of *TOM1L1* decreased with increasing pathological stage (Appendix A). In addition, survival analysis using the Kaplan–Meier plotter database validated that *CFB* and *PPP1R18* were significantly associated with poor OS in patients with ccRCC, whereas *TOM1L1* was significantly correlated with favorable OS (Appendix A). Furthermore, we conducted a pan-cancer analysis based on the Timer 2.0 database (Appendix A). Among the 19 cancer types, *PPP1R18*/*KIAA1949* (*p* = 2.98 × 10^−32^, Figure 9J) and *TOM1L1* (*p* = 3.87 × 10^−30^, Figure 9L) exhibited the highest specificity for KIRC/ccRCC, and *CFB* also presented excellent specificity for KIRC/ccRCC (*p* = 3.76 × 10^−13^, Figure 9K).

### 3.14. GSEA of the Key DMA-RMSG-Based Risk Score Model

To identify the potential molecular mechanism of the prognostic model, GSEA was performed between the different risk groups. The results showed that the pathways in low-risk patients with ccRCC were mostly related to mast cell immunity and carcinoma, including the vascular endothelial growth factor (VEGF) signaling pathway, TGF beta signaling pathway, proteasome, cytokine receptor interaction, and RCC (Figure 9M). Interestingly, most of the gene sets enriched in the high-risk group were correlated with autophagy, such as regulation of autophagy, the MTOR signaling pathway, PPAR signaling pathway, snare interactions in vesicular transport, and cell cycle (Figure 9N).

### 3.15. Relationship between the Prognostic Panel and RMC Infiltration

The characteristics of key RMCs varied between high- and low-risk patients with ccRCC. The raincloud plot showed that the infiltration scores of the RMCs in the low-risk group were significantly higher than those in the high-risk group (*p* < 0.0001, Figure 10A). Furthermore, the infiltration scores of RMCs significantly decreased with elevated risk scores (*R* = −0.4, *p* < 2.2 × 10^−16^), further demonstrating the crucial prognostic value of RMCs (Figure 10B). Patients were separated into four clusters according to their risk score and RMC infiltration. We observed that the group with high RMC infiltration and low risk scores had the best survival rate (*p* < 0.001, Figure 10C).

### 3.16. Difference in the TMB between the High- and Low-Risk Groups

TMB, a useful biomarker across many types of cancer, has been used to identify patients benefiting from immunotherapy. Therefore, genetic alterations in patients with ccRCC were investigated. The waterfall plot indicated that VHL had the highest alteration frequency in the TCGA ccRCC dataset (Figure 10D). Moreover, the TMB in the high-risk group was significantly higher (Figure 10E). The TMB of patients with ccRCC significantly increased with elevated risk scores (*R* = 0.14, *p* = 0.0066, Figure 10F). Furthermore, the survival rate in the low-TMB group was significantly better (*p* < 0.001, Figure 10G). After classifying patients with ccRCC into four subgroups based on their risk scores and TMB, we found that the combination of TMB and risk scores had greater prognostic value than TMB alone, with the best prognosis associated with low TMB and low risk scores (*p* < 0.001, Figure 10H).

### 3.17. Correlation between the Risk Score Model and Immune Checkpoints

To explore the underlying role of the prognostic panel in immunotherapy responsiveness, we analyzed the expression patterns of key immune checkpoint molecules, such as PD-1 and its ligands (PD-L1 and PD-L2), CTLA-4, lymphocyte activation gene 3 (LAG3), T-cell immunoreceptor with Ig and ITIM domains (TIGIT), inducible costimulator (ICOS), B- and T-lymphocyte-associated (BTLA), B7-H3, and CD28. The results indicated that the risk scores were positively related to the expression of PD-1, PD-L2, CTLA-4, LAG3, TIGIT, ICOS, BTLA, B7-H, and CD28 and negatively correlated with PD-L1 expression (*p* < 0.05, Figure 10I). Moreover, we found a negative correlation between RMCs and other risk factors (risk score, TMB, and most immune checkpoints), further indicating a protective effect of RMCs produced in ccRCC (Figure 10J).

### 3.18. Relationship between the DMA-RMSG-Based Prognostic Panel and ATGs

The GSEA result revealed that dysfunction of autophagy-associated pathways might also play a role in ccRCC distant metastasis. Therefore, we further investigated the potential autophagy-related targets that RMCs might regulate based on 222 ATGs (Appendix A). The results revealed that RGS19 exhibited the highest correlation coefficient with the risk scores (*R* = 0.73, *p* = 2.00 × 10^−103^, Figure 10K). We also explored the relationship between each key DMA-RMSG and *RGS19*. Consistent with previous results, *CFB* (*R* = 0.54, *p* = 5.27 × 10^−48^, Figure 10L) and *PPP1R18* (*R* = 0.74, *p* = 2.23 × 10^−105^, Figure 10M) were positively associated with *RGS19*, whereas *TOM1L1* (*R* = −0.61, *p* = 1.46 × 10^−64^, Figure 10N) was negatively associated with RGS19.

### 3.19. Construction and Validation of the Nomogram

To identify independent prognostic factors in ccRCC, we conducted univariate and multivariate Cox regression analyses on multiple clinicopathological variables. The results indicated that age, risk score, pathological stage, histological grade, N stage, and distant metastasis were significantly associated with poor prognosis in patients with ccRCC (*p* < 0.001, Figure 11A). More importantly, age, stage, risk score, and distant metastasis were screened as prognostic indicators independent of other clinical phenotypes (*p* < 0.05, Figure 11B). Taking the four independent prognostic factors together, a nomogram was established in the TCGA ccRCC cohort (Figure 11C). The calibration diagram exhibited good concordance between the nomogram-predicted OS and the observed OS after 1, 3, and 5 years (Figure 11D). Furthermore, ROC analysis showed that the AUC of the nomogram for 1-, 3-, and 5-year OS were 0.872, 0.805, and 0.771, respectively, demonstrating the excellent predictive performance of the nomogram (Figure 11E).

### 3.20. Implication of the Risk Score in Immunotherapy

To evaluate the relationship between the prognostic panel and immunotherapy response, ICI therapy scores of patients with ccRCC were calculated. In the TCIA cohort, compared with the low-risk group, the immunotherapy scores in high-risk patients were significantly increased in the PD-1- (*p* = 0.002, Figure 11G) and CTLA-4-positive treatment groups (*p* = 0.036, Figure 11H), indicating that high-risk patients were more sensitive to anti-PD-1 and anti-CTLA-4 immunotherapy. Furthermore, high-risk patients exhibited excellent sensitivity to the combination of anti-PD-1 and anti-CTLA-4 immunotherapy (*p* < 0.001, Figure 11I), revealing the potential immunotherapy regimen for patients with advanced ccRCC. However, no significant difference was observed between the high- and low-risk patients in PD-1- and CTLA-4-negative treatment groups (*p* = 0.953, Figure 11F). In the IMvigor210 cohort, low-risk patients were more sensitive to anti-PD-L1 therapy than high-risk patients (Figure 11J). Furthermore, compared with patients with complete or partial response, risk scores were significantly elevated in patients with progressive or stable disease (*p* = 0.018, Figure 11K). These results are in agreement with our previous finding (Figure 10I) that PD-1 and CTLA-4 were highly expressed in the high-risk group, whereas PD-L1 was highly expressed in the low-risk group.

### 3.21. Experimental Validation of the Three-Gene Signature Associated with Distant Metastasis

After rigorous screening, 33 paired ccRCC/normal tissue specimens were ultimately included in the current study. The detailed clinicopathologic information of the patients is shown in Appendix A. IHC was used to verify the expression levels of the three key DMA-RMSGs. In accordance with our previous findings of the database analysis, CFB and PPP1R18 had higher staining intensity in ccRCC tissues compared to normal kidney tissues (Figure 12A,B), whereas TOM1L1 had lower staining intensity in ccRCC tissues (Figure 12C). We also assessed the quantity of the positive strain in the IHC. The results showed that the positive cells of CFB and PPP1R18 were significantly higher in ccRCC tissues (*p* < 0.01, Figure 12G,I), whereas those of TOM1L1 were significantly lower (*p* < 0.01, Figure 12K). Compared with the nondistant metastasis samples, distant metastasis samples showed higher protein levels of CFB and PPP1R18 (Figure 12D,E) but a lower level of TOM1L1 (Figure 12F). Furthermore, compared to nondistant metastasis samples, CFB and PPP1R18 had a greater proportion of positive cells in distant metastasis samples (*p* < 0.01, Figure 12H,J), whereas TOM1L1 had fewer (*p* < 0.01, Figure 12L).

## 4. Discussion

Patients with advanced ccRCC have an adverse prognosis and limited therapeutic choices [5]. Accumulating evidence suggests that immunotherapy plays a non-negligible role in the treatment of patients with ccRCC with distant metastases [4,29]. Characterization of the composition and cellular states of TIICs helps to elucidate how the immune system might contribute to the immunotherapy response in advanced ccRCC [30]. Furthermore, screening of biomarkers is key to select better treatments, reduce costs, and enhance the outcomes of advanced ccRCC [31]. Although some studies have constructed immune-related predictive models that could influence ccRCC progression or prognosis [18,19,32], a risk signature based on distant metastasis-associated TIICs in ccRCC is still lacking.

As a critical component of the TME, TIICs affect the progression, prognosis, and treatment of various cancers. For example, tumor-infiltrating macrophages have been reported to blunt the antitumor response in glioblastoma by secreting TGF-β and IL-10 and recruiting regulatory T (Treg) cells [33]. In vulvar squamous cell carcinoma, subtypes of cytotoxic lymphocytes and NK cells infiltrating cancer nests are associated with patient prognosis [34]. Byrne et al. [35] suggested that patients with advanced breast cancer with high levels of tissue-resident memory T cells have elevated response rates to anti-PD-1 therapy. However, the roles of TIICs in advanced ccRCC were not fully understood until recently. Here, we assessed the distinct expression patterns of TIICs in distant and nondistant metastatic tissues. After evaluating the prognostic power of TIICs, we identified RMCs as the key TIICs that significantly inhibit distant metastasis and benefit OS and PFS in ccRCC.

Tumor-infiltrating mast cells (TIMCs) derived from myeloid stem cells activate immune cells and induce inflammation in the TME [36]. TIMCs exert protumorigenic or antitumorigenic effects according to the type of carcinoma, degree of tumor progression, and location of TIMCs in tumor tissues [37]. RMCs, an important type of TIMCs, are distinguished by their high content of electron-dense material evenly distributed throughout the major part of each granule in the cytoplasm [38]. RMCs are particularly important to the progression and treatment of various cancers [39]. For instance, a high RMC density is associated with a favorable prognosis in hepatocellular carcinoma [40]. However, Xie et al. [41] suggested that the distribution of RMCs in meningioma tissues is significantly higher than that in normal controls. Currently, studies on the role of RMCs in ccRCC, especially metastatic ccRCC are limited. Pan et al. [42] demonstrated that RMCs can act as protective factors for patients with ccRCC; however, a model underlying RMCs has not been constructed. Although Cui et al. [43] suggested that the infiltration of RMCs was negatively linked with the histological grade of ccRCC, the significance of RMCs in ccRCC metastasis has not been investigated. Here, based on the identified key RMCs, we screened RMSGs using WGCNA. Functional analysis revealed that RMSGs were significantly enriched in immune- and inflammation-related pathways, revealing that the immune system and inflammation may be involved in RMC-related distant metastasis in ccRCC. Compared to other studies that merely utilized bulk RNA-seq to filter key genes in metastatic ccRCC [44,45], ignoring the complex interaction of tumor cells and the associated nontumor constituents in TME, we identified ccRCC distant metastatic marker genes after systematic analysis of scRNA-seq data. Then, 12 prognosis-related DMA-RMSGs were screened by univariate Cox regression analysis and validated by KM prognosis analysis.

With the development of immunotherapy, it is crucial to screen the most sensitive group to improve immunotherapy response and prognosis in ccRCC. To achieve individual treatment, we categorized ccRCC patients into two immune clusters according to 12 prognosis-related DMA-RMSGs. According to the survival analysis, the ccRCC patients in subtype one had a worse OS and PFS. Furthermore, malignant phenotypes, such as distant metastasis, higher grade, higher stage, and higher TMB, were enriched in patients in cluster one. Subsequently, by conducting a series of rigorous analyses, three independent prognostic factors (*TOM1L1*, *CFB*, and *PPP1R18*) related to distant metastasis in ccRCC were identified.

The three key DMA-RMSGs play a non-negligible role in different cancers. Shimazaki et al. [46] suggested that high stromal *CFB* expression is associated with unfavorable clinical outcomes in patients with pancreatic ductal adenocarcinoma after surgery. Wu et al. [47] found that the expression of *CFB* in patients with N1 stage thyroid carcinoma was higher than that in those with N0 stage disease. However, there are very few reports on CFB in RCC. Cooley et al. [48] suggested that patients with RCC with increased *CFB* levels have faster disease progression and lower survival rates than those with decreased *CFB* levels. Furthermore, there are no relevant reports on the utilization of *CFB* expression in ccRCC. Studies on *PPP1R18* in human cancers are also limited. Meng et al. [49] reported that *PPP1R18* activates *LCN2* expression, which is related to esophageal cancer invasion. *PPP1R18* is highly expressed in ccRCC cell lines and tissues [50]. To the best of our knowledge, the roles of *PPP1R18* in distant metastatic ccRCC are still unknown. Chevalier et al. [51] reported that *TOM1L1* promotes *ERBB2*-induced breast cancer cell invasion. In cutaneous squamous cell carcinoma, *TOM1L1* expression levels are decreased and related to precursor lesions [52]. However, there have been no related studies on *TOM1L1* in kidney cancer. All of these studies further support the accuracy of our analyses of the three key DMA-RMSGs and the value of the current study in advanced ccRCC. Taken together, the three key DMA-RMSGs were utilized to construct a prognostic model in the TCGA dataset and validated in IGCG, GEPIA, and GEO cohorts, which yielded supportive results. Furthermore, IHC verified the independent protein expression levels of the three key DMA-RMSGs in ccRCC. After dividing patients with ccRCC into high- and low-risk groups, we found that patients in the low-risk group had better prognosis. Furthermore, we discovered that the risk model was closely associated with various clinical characteristics, indicating that it might be utilized as a useful index to predict clinical outcomes in patients with ccRCC. In addition, compared with the other clinical characteristics of patients with ccRCC, risk score had an independent influence on OS. In contrast to other studies that simply used common clinical variables to build nomograms in ccRCC [53,54], we integrated independent prognostic factors to construct the nomogram, improving the significance of the predictive model.

To investigate the possible mechanism by which RMCs regulate distant metastasis in ccRCC, we performed GSEA of the RMC-based signature. The analysis revealed that mast cell immune- and carcinoma-associated pathways were most abundant in the low-risk group, including the VEGF signaling pathway [55], TGF beta signaling pathway [56], proteasome [57], cytokine–cytokine receptor interaction [58], and RCC, whereas autophagy-associated pathways were mainly enriched in the high-risk group. These results further demonstrate that RMCs may play a crucial role in distant metastasis and the prognosis of ccRCC, which could be achieved by regulating immune function and autophagy. Substantial evidence has suggested that TMB has become an emerging biomarker of immunotherapy response among multiple solid tumors [59]. To further evaluate the application of our risk signature in immunotherapy, we analyzed RMC infiltration and TMB. In our analysis, patients with high infiltration of RMCs and low risk scores exhibited an excellent survival rate, further validating the prognostic role of RMCs in ccRCC. Furthermore, patients with low TMB and low risk scores exhibited the best prognosis, revealing the predictive value of the combination of TMB and risk signature in ccRCC.

Tumor immune escape is a crucial step in avoiding antitumor immune responses during cancer metastasis [60]. During tumor immune escape, cancer cells express high levels of immune-inhibitory signaling proteins, represented by immune checkpoints. To explore the underlying effect of our model on the immune escape of ccRCC cells, we assessed the relationship between the RMC-based signature and ten immune checkpoints [61]. The results showed that PD-L1 was upregulated in low-risk patients, whereas all other nine immune checkpoints were upregulated in high-risk patients, which further explained the high distant metastasis rate and low survival rate in high-risk patients. Subsequently, we investigated the potential effect of the RMC-based signature in predicting the response to ICI immunotherapy. In accordance with the former results, high-risk patients were more sensitive to anti-PD-1/CTLA-4 immunotherapy, whereas low-risk patients had a better response rate to anti-PD-L1 immunotherapy. These results revealed that the signature based on RMCs could help to screen patients with ccRCC with overexpressed immune checkpoints who might therefore benefit from different ICI therapies. In addition, considering the possible role of autophagy in ccRCC distant metastasis, which is in agreement with the results of our previous study [62], we further discovered that RGS19 shared the highest correlation with the risk score, which was likely the key ATG regulated by the risk model. RGS19 codes a guanosine triphosphatase-activating protein related to autophagy dysregulation [63]. The upregulated expression of RGS19 was associated with poor prognosis of bladder cancer patients [64]. However, the role of RGS19 in the autophagy of ccRCC requires further investigation.

## 5. Conclusions

In conclusion, we identified RMCs as the key TIIC, exerting a crucial role in the distant metastasis and prognosis of ccRCC. Subsequently, three key DMA-RMSGs predicting distant metastasis and prognosis in patients with ccRCC were identified to construct a prognostic panel and validated using different datasets and IHC. Furthermore, two immune clusters exhibiting distinct immune, clinical, and prognosis heterogeneity were distinguished. More importantly, the prognostic panel closely related to RMCs, TMB, and immune checkpoints also exhibited excellent performance in stratifying patients according to their responses to anti-PD-1/CTLA-4 and anti-PD-L1 immunotherapy. The limitations of our study mainly lie in the need to further investigate the specific molecular mechanisms by which RMCs and the DMA-RMSG-based signature affect ccRCC distant metastasis and prognosis. Taken together, our prognostic panel may help doctors predict clinical outcomes and design individual immunotherapies for patients with ccRCC.

## Figures and Tables

**Figure 1 cells-12-00180-f001:**
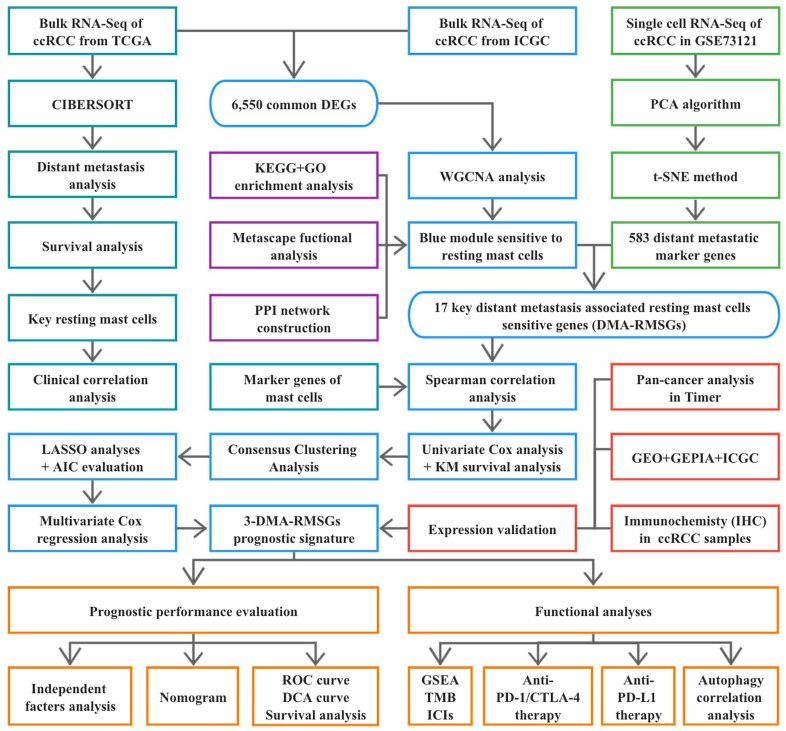
Workflow of the study. Turquoise rectangles represent the identification of key resting mast cells (RMCs). Green rectangles indicate screening of distant metastatic marker genes. Purple rectangles represent functional enrichment analysis of resting mast-cell-sensitive genes (RMSGs). Blue rectangles indicate the construction of the 3-DMA-RMSG prognostic panel. Red rectangles represent external validation of the 3-DMA-RMSG prognostic panel. Orange rectangles exhibit the predictive value and biological function of the 3-DMA-MRSG panel.

**Figure 2 cells-12-00180-f002:**
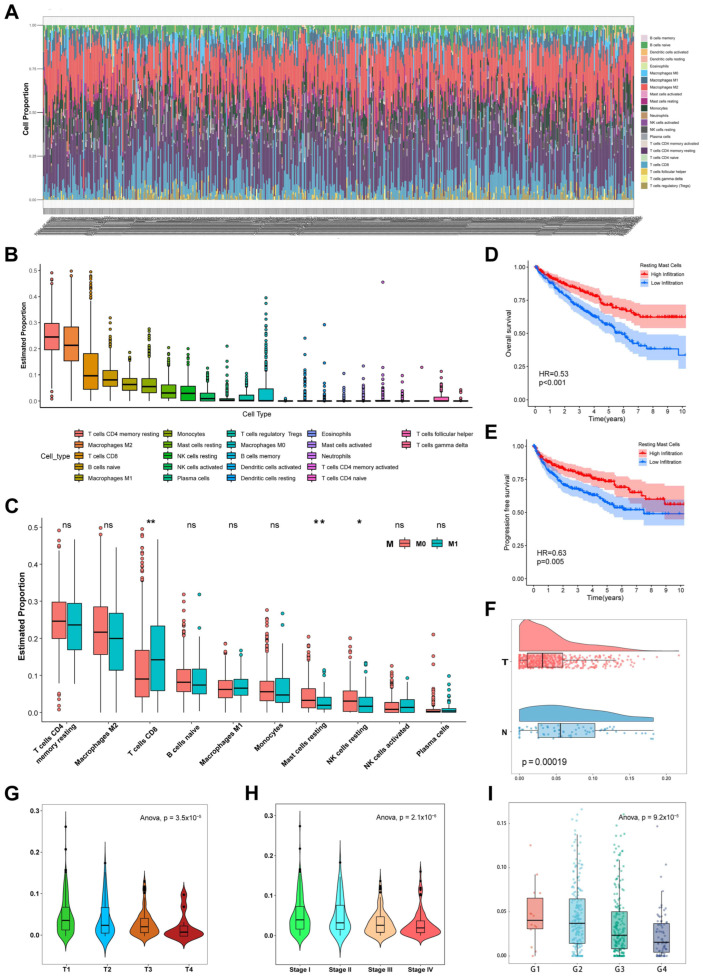
Identification and validation of the key TIICs. (**A**) The landscape depicts the proportions of TIICs in each ccRCC tissue. (**B**) Box plot describes the level of infiltration for the 22 TIICs. (**C**) Distribution of the top ten abundant TIICs between nondistant metastasis and distant metastasis ccRCC samples. Survival plots for OS (**D**) and PFS (**E**) display the prognostic value of RMCs. (**F**) Distribution of the number of RMCs between normal kidney tissues and ccRCC tissues. Violin plot visualizes the infiltration level of RMCs according to T stage (**G**), pathological stage (**H**), and histological grade (**I**). * *p* < 0.05, ** *p* < 0.01; ns, no significance.

**Figure 3 cells-12-00180-f003:**
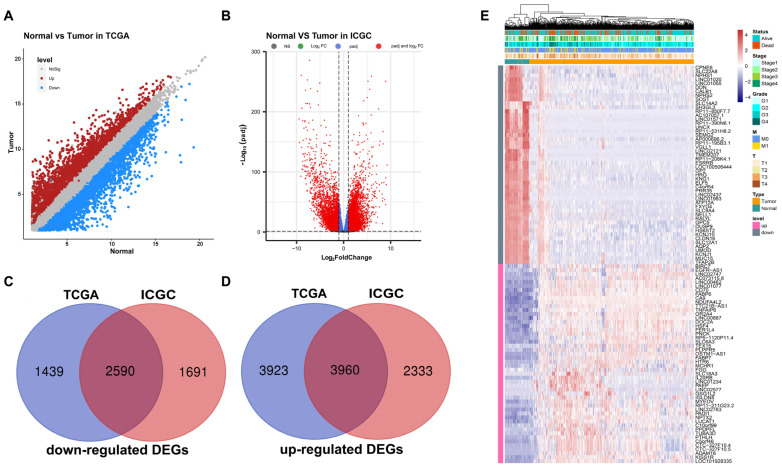
Screening and verification of DEGs. Volcano plots of DEGs between normal and ccRCC tissues based on TCGA (**A**) and ICGC datasets (**B**). Venn diagrams showing screened common downregulated (**C**) and upregulated DEGs (**D**). (**E**) Heat map of DEGs among several clinical phenotypes. DEGs, differentially expressed genes.

**Figure 4 cells-12-00180-f004:**
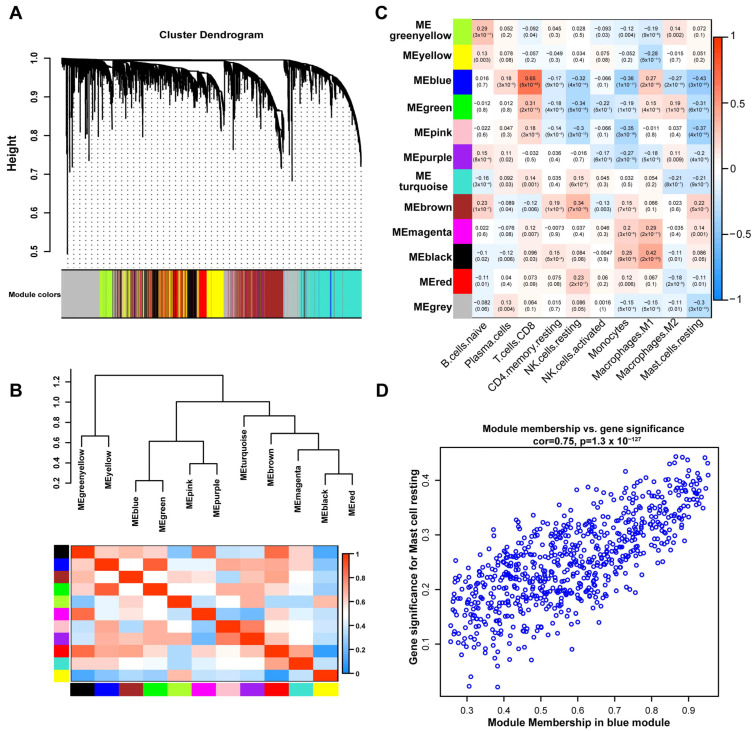
Identification and validation of the key modules closely related to RMCs. (**A**) Hierarchical clustering tree of DEGs based on the TCGA dataset. (**B**) Cluster dendrogram and heat map of the module eigengenes. (**C**) Heat map describing the relationship between the module eigengenes and the infiltration of the top ten TIICs. (**D**) Scatter plot of gene significance for RMCs vs. module membership in the key blue module.

**Figure 5 cells-12-00180-f005:**
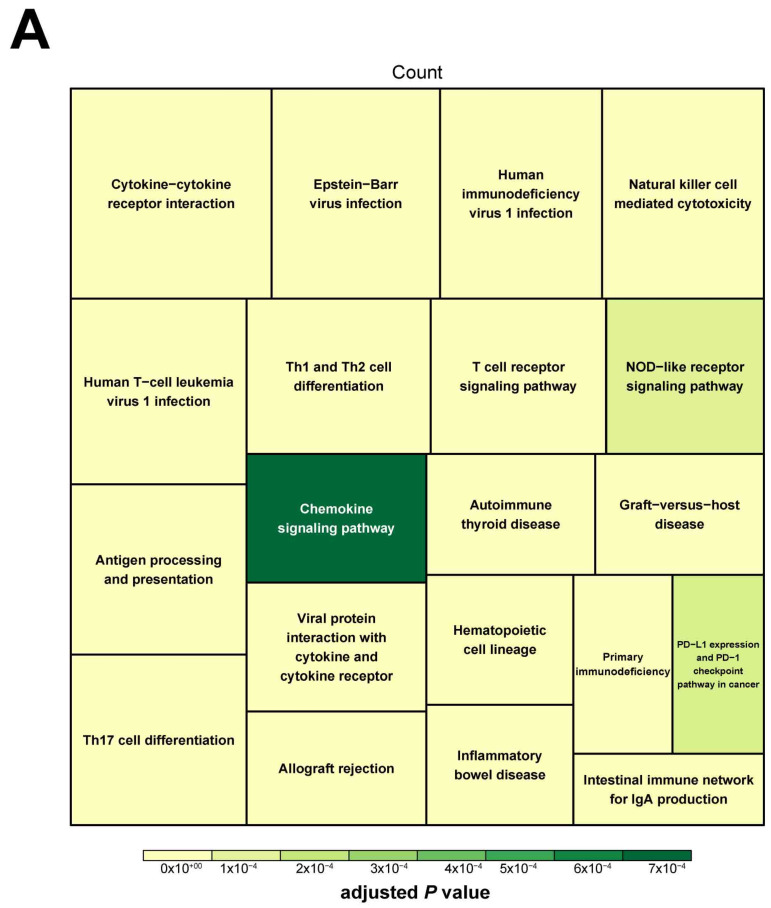
Functional enrichment analysis of RMSGs in the blue module. (**A**) Tree map of the top 20 most significant pathways in KEGG enrichment analysis. (**B**) The top 20 most significant GO terms and relevant genes in biological process are presented. (**C**) The chord diagram displays the top 20 most significant GO terms and involved genes in cell components. (**D**) The circle plot shows the top 20 most significant GO terms in molecular function. (**E**) Network interaction analysis of the signaling pathways using the Metascape database. (**F**) PPI network of RMSGs. RMSGs, resting mast-cell-sensitive genes; KEGG, *Kyoto Encyclopedia of Genes and Genomes*; GO, gene ontology; PPI, protein–protein interaction.

**Figure 6 cells-12-00180-f006:**
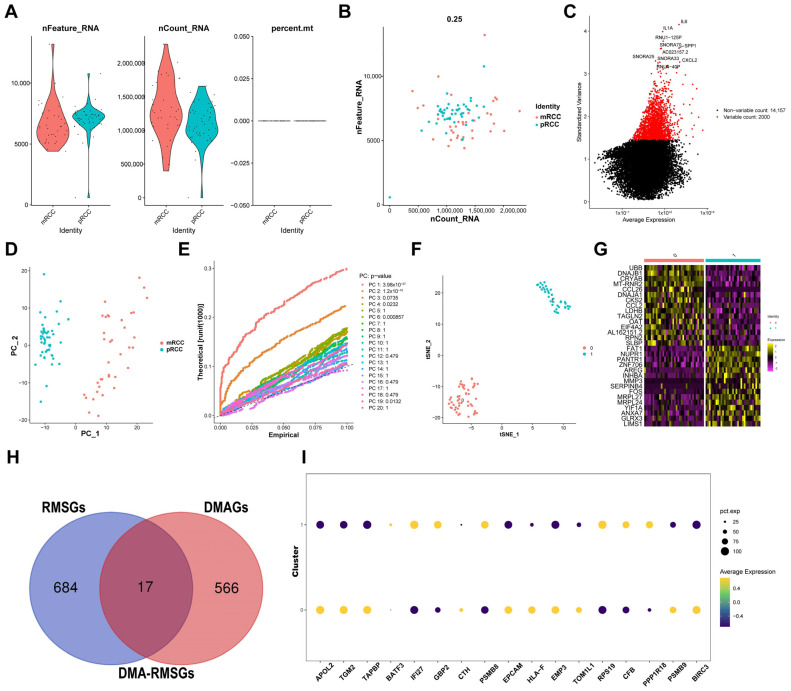
Single-cell RNA sequencing data in GSE73121 and DMA−RMSGs screening. (**A**) Quality control was performed to filter the ccRCC cells with poor quality. (**B**) Scatter plot shows a positive correlation between gene counts and sequencing depth. (**C**) Genes with high variability across cells were identified using a characteristic variance plot. (**D**) The two distinct distributions of ccRCC cell groups were obtained using the PCA method. (**E**) Significant principal components were screened using the estimated *p*-value. (**F**) ccRCC cells were further classified into two subgroups using the t-SNE algorithm. (**G**) Heat map displays the independent expression patterns of the top 30 DMAGs between the two subgroups. (**H**) Key DMA-RMSGs were obtained according to the intersection of RMSGs and DMAGs. (**I**) Dot plot presents the differential expression levels of the 17 DMA-RMSGs between the two ccRCC clusters.

**Figure 7 cells-12-00180-f007:**
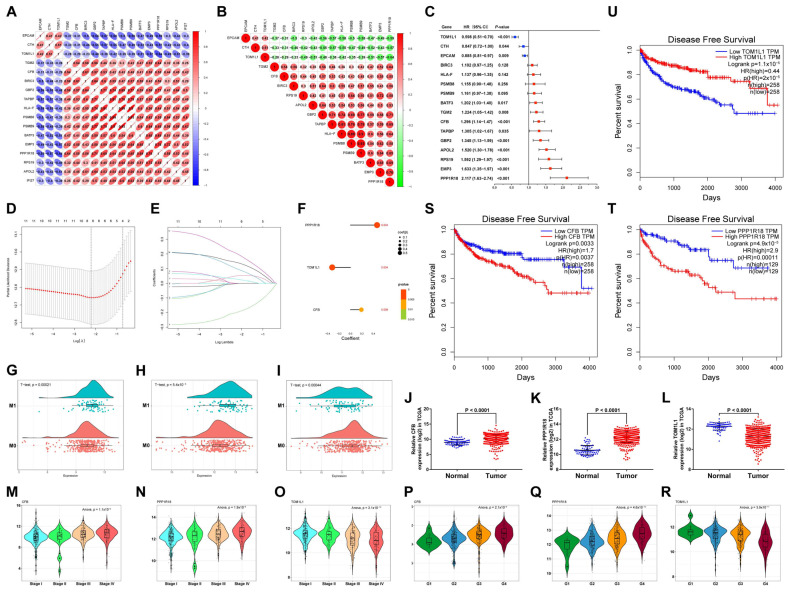
Identification and validation of the DMA-RMSGs that independently predict ccRCC prognosis. (**A**) Heat map of the correlations among the 17 DMA-RMSGs. (**B**) Heat map of the relationship among the 16 key DMA-RMSGs. (**C**) Univariate Cox regression analysis of the DMA-RMSGs. (**D**,**E**) LASSO regression analysis of the TCGA cohort. (**F**) Multivariate Cox regression analysis identified the coefficient of the three key DMA-RMSGs. Raincloud plots display the significantly distinct expression levels of *CFB* (**G**), *PPP1R18* (**H**), and *TOM1L1* (**I**) between nondistant and distant metastasis samples (*p* < 0.001). The independent expression patterns of *CFB* (**J**), *PPP1R18* (**K**), and *TOM1L1* (**L**) between normal and tumor tissues. The expression levels of *CFB* (**M**), *PPP1R18* (**N**), and *TOM1L1* (**O**) were significantly associated with the pathological stage (*p* < 0.001). Expression patterns of *CFB* (**P**), *PPP1R18* (**Q**), and *TOM1L1* (**R**) were significantly linked with the histological grade (*p* < 0.001). Survival analysis of DFS for *CFB* (**S**), *PPP1R18* (**T**), and *TOM1L1* (**U**).

**Figure 8 cells-12-00180-f008:**
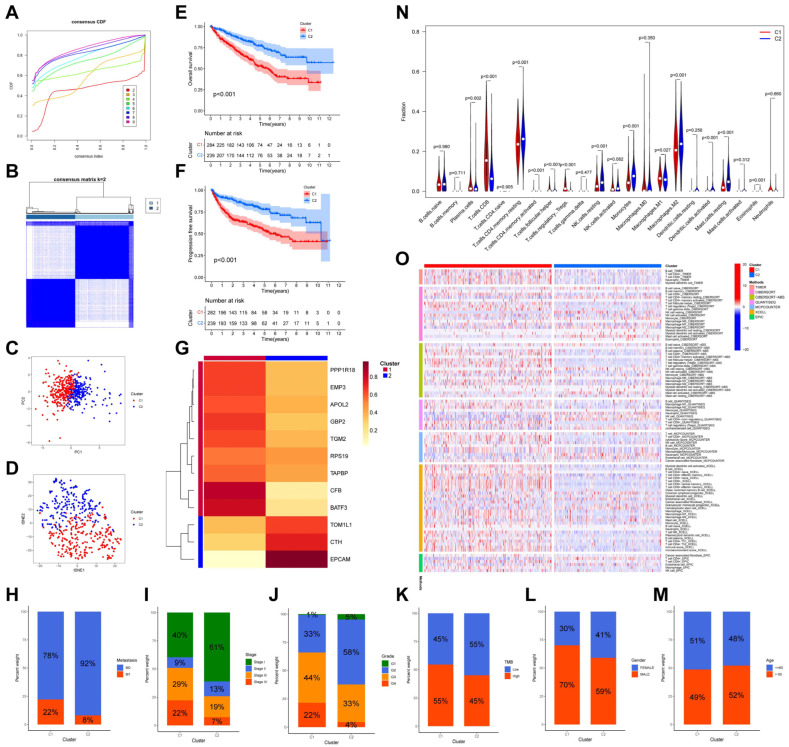
Identification and verification of novel immune subtypes in ccRCC. (**A**) The optimal k value for unsupervised clustering was determined according to the cumulative distribution function curve. (**B**) Sample clustering heat map exhibiting the consensus matrix in ccRCC. Stratification into two independent clusters verified by the PCA algorithm (**C**) and t−SNE algorithm (**D**). Kaplan–Meier curves for OS (**E**) and progression−free survival (**F**) between two ccRCC subtypes. (**G**) Clustering heat map representing the relationship between 2 subtypes and 12 prognosis−related DMA−RMSGs. Distribution of C1 and C2 among metastasis (**H**), stage (**I**), grade (**J**), tumor mutation burden (**K**), gender (**L**), and age (**M**) in ccRCC. Differential infiltration fractions of 22 TIICs between the two ccRCC clusters shown in violin plots (**N**) and a heat map (**O**).

**Figure 9 cells-12-00180-f009:**
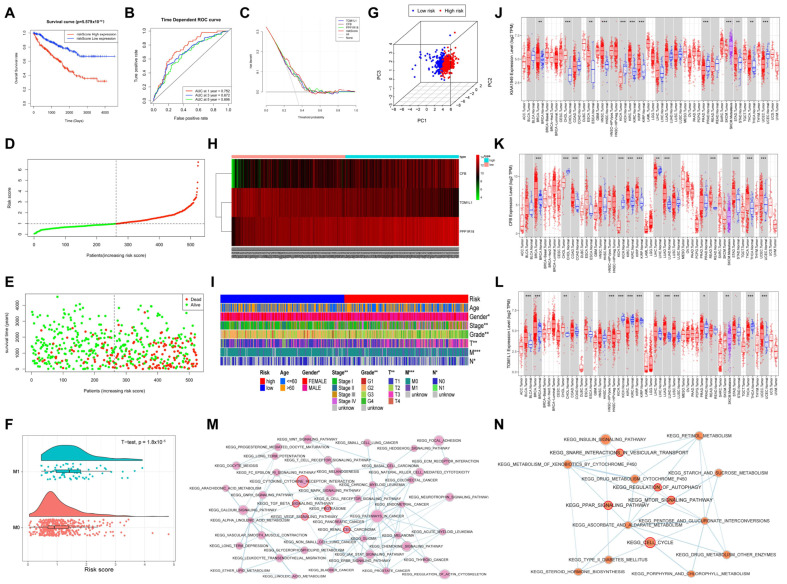
Evaluation and investigation of the prognostic panel. (**A**) OS analysis in the high- and low-risk groups. Time-dependent ROC curve (**B**) and DCA curve (**C**) of the risk score model. (**D**) Distribution of the risk scores of each patient with ccRCC in the TCGA set. (**E**) Survival status of patients with ccRCC with elevated risk scores. (**F**) Risk scores of patients with ccRCC were significantly correlated with distant metastasis (*p* < 0.001). (**G**) The independent expression distribution of the high- and low-risk groups was assessed using principal component analysis. (**H**) Heat map depicts the differential expression of the three key DMA-RMSGs between high- and low-risk patients. (**I**) The relationship between clinical traits and risk scores in TCGA. Pan-cancer analysis of *KIAA1949/PPP1R18* (**J**), *CFB* (**K**), and *TOM1L1* (**L**). The GSEA network revealed the pathways enriched in low-risk patients (**M**) and high-risk patients (**N**). Representative gene sets are labeled with a red outline. *p* < 0.05 was considered statistically significant. ROC, receiver operating characteristic; DCA, decision curve analysis. * *p* < 0.05, ** *p* < 0.01, *** *p* < 0.001.

**Figure 10 cells-12-00180-f010:**
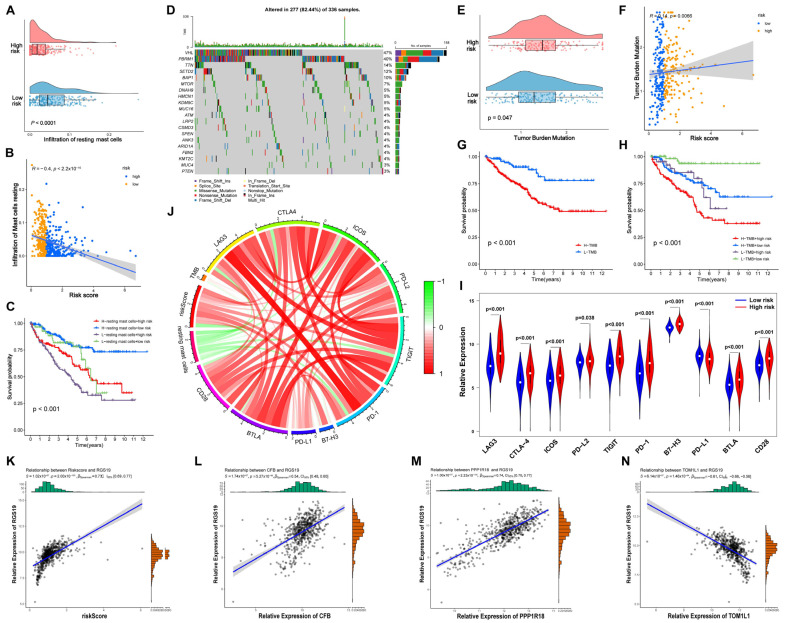
Relationship of the prognostic panel with immune-predicting factors and autophagy-related genes. (**A**,**B**) The risk model was significantly negatively linked with the infiltration of RMCs (*p* < 0.001). (**C**) The integration of risk score and RMC infiltration presents a great prognostic value (*p* < 0.001). (**D**) Waterfall plot illustrates the highly mutated genes. (**E**,**F**) The risk signature was significantly positively related to the TMB (*p* < 0.001). (**G**) OS curves of the high- and low-TMB patients. (**H**) The combination of the risk score and the TMB presents a significant prognostic value (*p* < 0.001). (**I**) The distinct expression of representative immune checkpoint genes in high- and low-risk patients. (**J**) Circle diagram exhibits the correlation among the prognostic panel, RMCs, TMB, and immune checkpoint genes. Scatter plot displays the relationship between *RGS19* and the risk score (**K**), *CFB* (**L**), *PPP1R18* (**M**), and *TOM1L1* expression (**N**). *p* < 0.05 was considered statistically significant. TMB, tumor mutation burden.

**Figure 11 cells-12-00180-f011:**
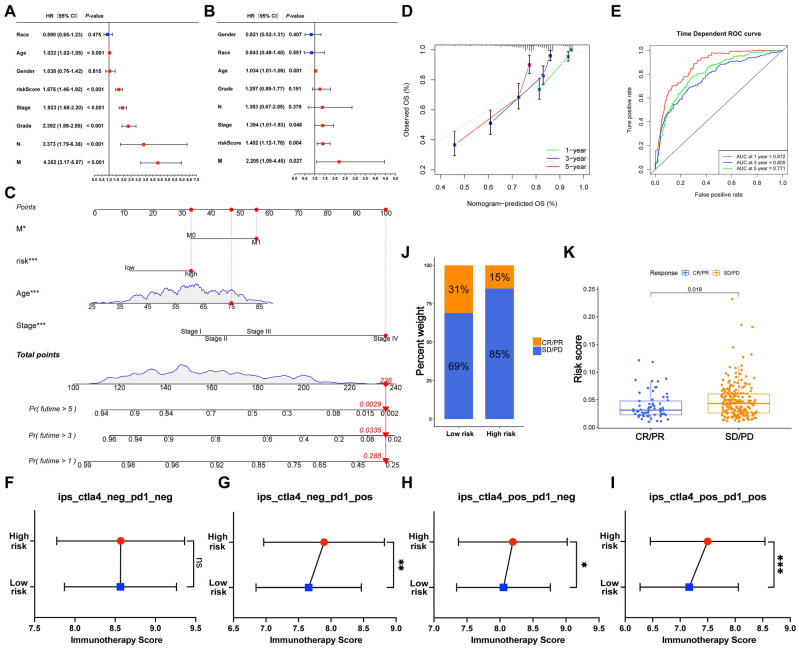
Construction of the nomogram and the predictive power of the RMC-based signature in immunotherapy response. Univariate (**A**) and multivariate (**B**) Cox regression analyses identified the pathological stage, M stage, age, and risk score as independent prognostic factors. (**C**) A nomogram was established according to the selected independent prognostic factors. The calibration diagram (**D**) and ROC curve (**E**) show the performance of the nomogram in predicting prognosis after 1, 3, and 5 years. (**F**–**I**) The differing sensitivity of the high- and low-risk patients subjected to anti-PD-1 and anti-CTLA-4 immunotherapy. (**J**) The proportions of patients who responded to anti-PD-L1 immunotherapy in the high- and low-risk groups in the IMvigor210 dataset. CR, complete response; PR, partial response; SD, stable disease; PD, progressive disease. (**K**) The ability of the risk score model to predict anti-PD-L1 immunotherapy response. * *p* < 0.05, ** *p* < 0.01, *** *p* < 0.001; ns, no significance.

**Figure 12 cells-12-00180-f012:**
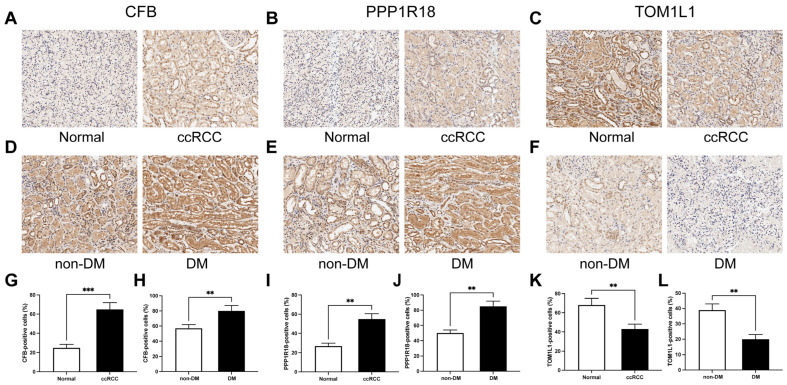
Validation of the expression of three key DMA-RMSGs using IHC. The differential protein levels of CFB (**A**), PPP1R18 (**B**), and TOM1L1 (**C**) between normal and ccRCC tissues. IHC staining revealed the distinct expression patterns of CFB (**D**), PPP1R18 (**E**), and TOM1L1 (**F**) between the nondistant metastasis and distant metastasis samples. The positive cells of CFB (**G**), PPP1R18 (**I**), and TOM1L1 (**K**) between normal and ccRCC tissues. The positive cells of CFB (**H**), PPP1R18 (**J**), and TOM1L1 (**L**) between the nondistant metastasis and distant metastasis samples. ** *p* < 0.01, *** *p* < 0.001; ns, no significance. DMA-RMSGs, distant metastasis-associated resting mast-cell-sensitive genes; IHC, immunohistochemistry.

## Data Availability

Publicly available datasets were analyzed in this study. The dataset used in this study was acquired from the open-source databases TCGA, ICGC, and GEO. TCGA: (http://cancergenome.nih.gov/, accessed on 15 January 2022); ICGC: (https://icgc.org/, accessed on 25 January 2022); GEO: GSE73121, GSE53757, GSE66272, and GSE126964 were downloaded from the GEO database (http://www.ncbi.nlm.nih.gov/geo/, accessed on 12 March 2022).

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
