# Peer review of "Identification and Validation of the Prognostic Panel in Clear Cell Renal Cell Carcinoma Based on Resting Mast Cells for Prediction of Distant Metastasis and Immunotherapy Response"

_cells, 2023, doi:10.3390/cells12010180_

Round 1

Reviewer 1 Report

This is an interesting research article, I have several suggestions to authors to improve the article.

1. This is a huge research article, I suggested to add abbreviation list to make it easier to read.

2. Figure 1 had no figure legends, please at least to explain the meaning of each color frame.

3. Figure 9, The three genes cannot distinguish between KIRC (Kidney renal clear cell carcinoma) and KIRP (Kidney renal papillary cell carcinoma). It’s very important to add more genes candidate like Figure 8G to find some different between KIRC and KIRP.

4. In line 651 or line 893 should be added some reference about the role RGS19 on autophagy.

5. It's needed to delete some blank in line 801-803.

Author Response

Manuscript ID: cells-2096047

Title: Identification and Validation of the Prognostic Panel in Clear Cell Renal Cell Carcinoma Based on Resting Mast Cells for Predicting Distant Metastasis and Immunotherapy Response

Dear editor,

We are truly grateful to the critical comments and thoughtful suggestions of reviewers. Based on these helpful comments and constructive advice, we have made careful modifications on the original manuscript. All changes made to the text are in yellow color. Below you will find our point-to-point corrections in the authors’ proof.

Responses to Reviewer #1

Point 1: This is a huge research article, I suggested to add abbreviation list to make it easier to read.

Response 1: Thank you very much for going through our paper carefully and your nice advice. We added abbreviation list to make it easier to read as you required (lines 844 to 852).

Point 2: Figure 1 had no figure legends, please at least to explain the meaning of each color frame.

Response 2: Thank you very much for going through our paper carefully and your constructive advice. As you suggested, we added figure legends to explain the meaning of each color frame in figure 1 (lines 107 to 113).

Point 3: Figure 9, The three genes cannot distinguish between KIRC (Kidney renal clear cell carcinoma) and KIRP (Kidney renal papillary cell carcinoma). It’s very important to add more genes candidate like Figure 8G to find some different between KIRC and KIRP.

Response 3: Thank you very much for going through our paper carefully and your nice advice. As you suggested, expression differences of normal and tumor tissues simultaneously exist in KIRC and KIRP. However, the difference of KIRC is greater than that of KIRP. As shown in Table S6, the difference of PPP1R18 in KIRC (2.98E-32) is about 1024 greater than that of KIRP (3.78E-06). The difference of TOM1L1 in KIRC (3.87E-30) is about 1022 greater than that of KIRP (4.37E-08). The difference of CFB in KIRC (3.76E-13) is about 10,000 greater than that of KIRP (2.79E-09). In addition, the main purpose of this article is to identify the biomarkers serving patients with ccRCC to predict their distant metastasis, prognosis and immunotherapy response.

Point 4: In line 651 or line 893 should be added some reference about the role RGS19 on autophagy.

Response 4: Thank you very much for going through our paper carefully and your constructive advice. We added some references about the role RGS19 on autophagy as you suggested (references 63 and 64).

Point 5: It's needed to delete some blank in line 801-803.

Response 5: Thank you very much for going through our paper carefully and your advice. we deleted the blank in line 801-803 as you suggested.

Reviewer 2 Report

The aim of the authors is relevant and intriguing but the paper is too long with repetitive concepts.

The bibliography should be updated regarding the therapeutic options (row 59, 745).

No biomarker proved to be useful in clinical practice, though active investigation is ongoing in this field and PD-L1 is unuseful. Actually we lack validated biomarkers able to drive our everydays clinical decisions.

RCC remains an heterogeneous disease, the characteristics of the primary tumor may differ from those of the metastatic lesions. Tumor microenvironment and RMCs may change during different stage of the disease.

Author Response

Manuscript ID: cells-2096047

Title: Identification and Validation of the Prognostic Panel in Clear Cell Renal Cell Carcinoma Based on Resting Mast Cells for Predicting Distant Metastasis and Immunotherapy Response

Dear editor,

We are truly grateful to the critical comments and thoughtful suggestions of reviewers. Based on these helpful comments and constructive advice, we have made careful modifications on the original manuscript. All changes made to the text are in yellow color. Below you will find our point-to-point corrections in the authors’ proof.

Responses to Reviewer #2
Point 1: The aim of the authors is relevant and intriguing but the paper is too long with repetitive concepts.

Response 1: Thank you very much for going through our paper carefully and your constructive advice. As you suggested, we deleted repetitive concepts (more than 1,000 words) in the manuscript, which mainly focus on the discussion and figure legend section. All changes in the manuscript are presented using revisions mode in Microsoft Word.

Point 2: The bibliography should be updated regarding the therapeutic options (row 59, 745).

Response 2: Thank you very much for going through our paper carefully and your nice advice. We updated the bibliography of the therapeutic options as you required (references 4, 5 and 8).

Point 3: No biomarker proved to be useful in clinical practice, though active investigation is ongoing in this field and PD-L1 is unuseful. Actually we lack validated biomarkers able to drive our everydays clinical decisions.

Response 3: Thank you very much for going through our paper carefully and your constructive advice. As you suggested, no biomarker proved to be useful in clinical practice of ccRCC currently. Considering the lack of validated biomarkers able to drive our everydays clinical decisions, we aim to identify the key biomarkers based on RMCs (resting mast cells) to predict distant metastasis, prognosis, progression and response to immunotherapy response in ccRCC. Furthermore, validation of external dataset and immunohistochemistry also showed good stability of the constructed gene panel. In addition, based on the identified prognosis-related biomarkers, we categorized ccRCC patients into two immune clusters with distinct immune, clinical and prognosis heterogeneity. Therefore, to some extent, our prognostic panel may help doctors predict clinical outcomes and design individual immunotherapies for patients with ccRCC.

Point 4: RCC remains an heterogeneous disease, the characteristics of the primary tumor may differ from those of the metastatic lesions. Tumor microenvironment and RMCs may change during different stage of the disease.

Response 4: Thank you very much for going through our paper carefully and your excellent view. As you suggested, the development of tumor is a dynamic evolutionary process. The main purpose of this study is to identified the key tumor-infiltrating immune cell and genes that play a role in the distant metastasis of ccRCC. However, considering the heterogeneity of primary tumors, we further found that the TIICs (tumor-infiltrating immune cells)/genes we screened also played a non-negligible role in the development of ccRCC, such as oncogenesis, T stage, pathological stage, histological grade and prognosis (Figures 2E-I and Figures 7G-U). In addition, considering the complex interaction between tumor cells and their microenvironments, we also conducted a single cell analysis to better understand the heterogeneity of tumor cells (Figure 6).

Round 2

Reviewer 1 Report

All suggestion had been improved. 

Reviewer 2 Report

None